# Blockade of Glial Connexin 43 Hemichannels Reduces Food Intake

**DOI:** 10.3390/cells9112387

**Published:** 2020-10-31

**Authors:** Florent Guillebaud, Manon Barbot, Rym Barbouche, Jean-Michel Brézun, Kevin Poirot, Flora Vasile, Bruno Lebrun, Nathalie Rouach, Michel Dallaporta, Stéphanie Gaige, Jean-Denis Troadec

**Affiliations:** 1Cognitive Neuroscience Laboratory, Aix-Marseille University, UMR 7291 CNRS, 13003 Marseille, France; florent.guillebaud@gmail.com (F.G.); manon.BARBOT@amu-amu.fr (M.B.); rym.barbouche@univ-amu.fr (R.B.); kevin.poirot@univ-amu.fr (K.P.); bruno.lebrun@univ-amu.fr (B.L.); michel.dallaporta@univ-amu.fr (M.D.); 2Institute of Movement Sciences, Aix-Marseille University, UMR 7287 CNRS, 13009 Marseille, France; jean-michel.brezun@univ-amu.fr; 3Neuroglial Interactions in Cerebral Physiopathology, Collège de France, CIRB CNRS 7241, Inserm U1050, Labex Memolife, PSL Research University, 75231 Paris, France; flora.vassile@college-de-france.fr (F.V.); nathalie.rouach@college-de-france.fr (N.R.)

**Keywords:** hypothalamus, brainstem, astrocytes, tanycytes, mimetic peptide

## Abstract

The metabolic syndrome, which comprises obesity and diabetes, is a major public health problem and the awareness of energy homeostasis control remains an important worldwide issue. The energy balance is finely regulated by the central nervous system (CNS), notably through neuronal networks, located in the hypothalamus and the dorsal vagal complex (DVC), which integrate nutritional, humoral and nervous information from the periphery. The glial cells’ contribution to these processes emerged few year ago. However, its underlying mechanism remains unclear. Glial connexin 43 hemichannels (Cx43 HCs) enable direct exchange with the extracellular space and can regulate neuronal network activity. In the present study, we sought to determine the possible involvement of glial Cx43 HCs in energy balance regulation. We here show that Cx43 is strongly expressed in the hypothalamus and DVC and is associated with glial cells. Remarkably, we observed a close apposition of Cx43 with synaptic elements in both the hypothalamus and DVC. Moreover, the expression of hypothalamic Cx43 mRNA and protein is modulated in response to fasting and diet-induced obesity. Functionally, we found that Cx43 HCs are largely open in the arcuate nucleus (ARC) from acute mice hypothalamic slices under basal condition, and significantly inhibited by TAT-GAP19, a mimetic peptide that specifically blocks Cx43 HCs activity. Moreover, intracerebroventricular (i.c.v.) TAT-GAP19 injection strongly decreased food intake, without further alteration of glycaemia, energy expenditures or locomotor activity. Using the immediate early gene c-Fos expression, we found that i.c.v. TAT-GAP19 injection induced neuronal activation in hypothalamic and brainstem nuclei dedicated to food intake regulation. Altogether, these results suggest a tonic delivery of orexigenic molecules associated with glial Cx43 HCs activity and a possible modulation of this tonus during fasting and obesity.

## 1. Introduction

The metabolic syndrome, which comprises obesity and diabetes, is a major public health problem and the comprehension of energy homeostasis control remains an important worldwide issue. The energy balance is finely regulated by the central nervous system (CNS), notably through neuronal networks, located in the hypothalamus and the DVC including the nucleus tractus solitarius (NTS), which integrate nutritional, humoral and nervous information from the periphery and in turn adjust energy expenditure and food intake. Decoding the precise mechanisms regulating energy balance remains a challenge necessary to efficiently treat metabolic disorders. Evidence showing contribution of glial cells in these processes are now mounting. Indeed, recent studies showed that glial cells in both the hypothalamus [1,2,3] and brainstem [4,5,6] interact with neurons and in turn finely tune energy homeostasis. However, the mechanisms of neuron-glia interactions in the context of food intake control remain to be deciphered. A typical feature of glial cells is their high expression level of Cx43, a protein able to form not only gap junction (GJ) channels, but also connexin 43 hemichannels (Cx43 HCs), in allowing the release of neuroactive substances.

Cxs proteins are highly conserved between species and are named according to their molecular weight, from 26 to 60 kDa. There are around 20 different genes (20 in mice and 21 in humans) coding for as many proteins [7]. The Cxs are the connexon base subunits, a hexamer forming a hydrophilic channel 2 nm in diameter. When two connexons of different cells join together, they form an intercellular channel and several thousands of such channels form a GJ [8]. In the central nervous system (CNS), glia and particularly astrocytes primarily and strongly express Cx43, Cx30 and to a lesser extent Cx26 [9]. These Cxs GJ contribute to a cytoplasmic continuity, providing the structural basis for an extensive astroglial network [10]. Alternatively, connexons not engaged in GJ form hemichannels (HCs), which are permeable to small molecules (less than 1.5 kDa) such as Ca^2+^, glucose, adenosine tri-phosphate (ATP), D-serine and glutamate [11]. The opening of these HCs is regulated by different factors such as Ca^2+^, pH, hormones [9]. The study of the functional role of these HCs is now possible thanks to the development of peptides specifically blocking their activity, such as TAT-GAP19, a mimetic peptide preventing interaction between the C-terminal part of Cx43 with the L2 loop [12]. HCs contribute to the release of various neuroactive molecules and can thus modulate of neuronal networks and associated behaviors [13,14,15,16].

The aim of the present study was to determine the possible contribution of glial HCs in rodent’s energy balance. For this, we have characterized Cx43 expression and its variations during nutritional status changes in mouse hypothalamus and brainstem. Next, we investigated the structural relationship between Cx43 and synaptic elements in these brain regions and studied the impact of Cx43 HCs on energy balance and activation of food-intake-dedicated neuronal networks using pharmacological inhibition.

## 2. Materials and Methods

### 2.1. Animals

Adult male C57BL/6J mice were purchased from Charles River Laboratories (L’Abresle, France). For TAT-GAP19 or mutated TAT-GAP19-I14A injections, C57BL/6J adult male mice (8 to 10 weeks old) were housed in standard cages, maintained in a controlled environment (12/12 h light-dark cycle, 22 °C and 40–50% humidity) with free access to water and food (AO4 pellets, SAFE, Augy, France). Some mice received a high fat diet (HFD, 60% fat; SSNIFF GmbH, Germany). Body weight and food intake were measured every week. In the HFD group, only mice which gained more than 30% of weight mass after 15 weeks compared to mice fed with the standard food (NC) were considered as obese and conserved for the tissue analyzes. Both obese and lean mice were sacrificed after 15 weeks of diet and tissues samples were collected for further analysis. All experiments were conformed to EC Council Directive (2010/63/UE) and the French “Direction Départementale de la Protection des Populations des Bouches-du-Rhône” (animal housing facility accreditation: N° D13 0556; animal experimentation accreditation: APAFIS#2207-2015100819154639 v4).

### 2.2. Peptides

TAT-GAP19 and TAT-GAP19-I14A were synthesized by Genopshere Biotechnologies using the following amino acid sequences: YGRKKRRQRRRKQIEIKKFK. The mutated and noneffective peptide was strictly identical except for Alanine at 14th position instead of Isoleucine (mutation I130A in Cx43 protein). TAT peptide was added to GAP19 to allow for its cells penetration. To detect these peptides using histology, we used the same sequences coupled with N-terminal biotin.

### 2.3. Surgery and Intracerebroventricular Injection

Cannula implantation was performed as previously described by Gaigé and colleagues [17]. One week post-surgery, mice were i.c.v. injected with 6 µL (2.5 µL/min) of physiological saline (NaCl 0.9%), TAT-GAP19 (12.5, 25 or 50 µg; 6 µL), or TAT-GAP19-I14A (25 µg) solutions at the beginning of the dark phase.

For acute injection of biotinylated TAT-GAP19 and TAT-GAP19-I14A, mice were anaesthetized by intraperitoneal (i.p.) injection of ketamine (100 mg/kg) and xylazine (16 mg/kg). Under stereotaxic control, mice were then i.c.v. injected with 6 µL (2.5 µL/min) of physiological saline (NaCl 0.9%), TAT-GAP19 (25 µg) or TAT-GAP19-I14A (25 µg) solutions. At different times after biotinylated mimetic peptides injection, anaesthetized animals were perfused intracardially with ice-cold 0.1M phosphate buffered saline (PBS, pH 7.4) and then with ice-cold, freshly prepared 4% paraformaldehyde (PFA) in 0.1M PB. The brains were immediately removed, post-fixed for 1h in 4% PFA at room temperature, rinsed overnight in PBS and then cryoprotected for 24–48 h in 30% sucrose at 4 °C. The brains were frozen in isopentane (−40 °C), and coronal sections (40 µm thick) of the brain were made with a cryostat (Leica CM3050, Leica Biosystems, Wetzlar, Germany) and collected serially in 0.1M PBS. Immunochemistry of biotin was then performed as described below. 

### 2.4. Food Intake Measurements

For food consumptions, ad libitum fed mice received i.c.v. injections of NaCl (0.9%), TAT-GAP19 (12.5 or 25 µg) or TAT-GAP19-I14A (25 µg) 30 min prior to lights-off. Immediately after treatment, a fresh supply of pre-weighed powdered food was given. Continuous food intake measurements and meal pattern analyses following i.c.v. injections were performed using a previously described feeding station [17]. Food intake was calculated as the difference between the pre-weighed and the remaining powder measured with a precision balance (0.01 g; Denver Instrument from Bioblock, San Diego, CA, USA). To analyze the meal pattern, we modified polypropylene cages (Cage S; Charles River Laboratories, Wilmington, MA, USA) with a feeding station allowing access to a food jar placed below the cage, and filled with powdered food (AO4). The feeding station was designed to avoid feces and urine dropping into the food jar and also to avoid food spillage. Analysis first proceeded through a MATLAB-developed software, filtering out high frequency oscillations in food jar weights and collecting the start time, end time and weight amplitude of the stepwise reductions in food jar weights. Analyzed data were exported to Excel for further analysis. Meals were defined by stepwise reductions in food jar weight of at least 0.02 g in amplitude (i.e., twice the balance precision) and 30 s in duration.

### 2.5. Indirect Calorimetry and Locomotor Activity

Indirect calorimetry recording was performed with an indirect open-circuit calorimeter Oxylet Physiocage system (Panlab, Cornella, Spain), equipped with a laser absorption O_2_ sensor, an infrared technology CO_2_ sensor, and weight transducers for the continuous assessment of the food and water consumption. Room air flowed through each chamber at a rate of 450 mL/min. The O_2_ and CO_2_ levels were measured during 3 min sampling periods every 30 min and data were analyzed with the METABOLISM software. Oxygen consumption (VO_2_), and carbon dioxide production (VCO_2_) were expressed in mL/min/kg^0.75^. The respiratory quotient (RQ) was determined by the ratio V_CO2_/V_O2_ and energy expenditure (EE) was calculated according to the formula EE (kcal/day/kg^0.75^) = VO_2_ _ 1.44 _ [3.815 + (1.232 _ RQ)]. Mice were habituated to the metabolic chambers for 24 h. The O_2_/CO_2_ analyzer was calibrated with purified gas standards and the food hoppers replenished every day at the beginning of the light phase. TAT-GAP19 (50 µg) or vehicle was administered just before the dark phase, on day 2 of a 2-day recording session. The mean values for V_O2_, V_CO2_, respiratory quotient (RQ) and energy expenditure (EE) were compared for each group between the dark phase before and after treatment. To measure locomotor activity, cages were kept on weight transducers allowing the detection of animal movements (1 cumulative measurement per 30 min period, Arbitrary Unit, AU).

### 2.6. Oral Glucose Tolerance Test and Glycaemia Measurements

Glycaemia measurements were acquired by collecting blood samples from THE tail vein. A section of tail vein was pierced by a needle, and blood was directly applied to a strip for glucose detection using a glucose meter (Accu-Chek Performa nano, Roche Diagnostics Corporation, Indianapolis, IN, USA). Glycaemia was determined before the TAT-GAP19 injection and then at 10, 20, 30, 60, 90, 120, 180 min after its injection. For oral glucose tolerance test, i.c.v. TAT-GAP19 treatment or vehicle was performed 30 min before animals received a bolus of glucose per os (1.5 g/kg). Glycaemia was then determined before glucose administration and 5, 10, 20, 30, 60, 90, 120 and 180 min after its administration.

### 2.7. Tissue Collection

Mice were anaesthetized using i.p. injection of ketamine (100 mg/kg) and xylazine (16 mg/kg) and sacrificed by decapitation. The prefrontal cortex, hypothalamus and DVC were collected, rinsed in saline solution (NaCl 0.9%), snap-frozen in liquid nitrogen and stored at ‒80 °C until further analysis.

### 2.8. Real-Time PCR Analysis 

Animals used for reverse transcriptase polymerase chain reaction (RT-PCR) analysis were sacrificed at the end of the treatment period. mRNA extraction and RT were made as described previously [17]. Briefly, total RNAs were extracted from frozen organs using TRI Reagent^®^ (Sigma-Aldrich, St. Louis, MA, USA) according to the manufacturer’s instructions. RT was realized using Moloney Murine Leukemia Virus Reverse Transcriptase in the presence of random hexamer primers (Promega, Madison, WI, USA). Gene expression analysis by real time PCR was performed using the LightCycler^®^ 480 System (Roche Applied Science, Penzberg, Germany). The equivalent of 40 ng initial RNAs were subjected to PCR amplification in a 10 μL final volume using specific primers 0.25 μM (Table 1) and KAPA SYBR^®^ FAST Master Mix (2X) optimized for Roche LightCycler^®^ 480 (CliniSciences, KAPABiosystems, Wilmington, MA, USA). The generation of specific PCR products was confirmed by melting-curve analysis. Glyceraldehyde-3-phosphate dehydrogenase gene (GAPDH) was used as internal reference gene. 

### 2.9. Western Blotting

Hypothalamus tissues were homogenized in Radioimmunoprecipitation Assay (RIPA) buffer (50 mM Tris pH 8.0, 0.1% sodium dodecyl sulfate, 1% Triton X- 100, 0.5% sodium deoxycholate, 150 mM NaCl, and 1 mM EDTA) supplemented with protease and phosphatase inhibitors cocktail (bimake.com, UK) then maintained under constant agitation in ice for 1 h. Extracts were centrifuged at 12,000× *g* for 20 min at 4 °C to remove tissue debris. The supernatants were collected for assessment of protein concentration by Bicinchoninic Acid (BCA) Protein Assay method (Novagen, Germany). Then equivalent amounts of protein (30 μg) were separated on 12% SDS-PAGE gel and transferred onto nitrocellulose membranes (Amersham, Germany) to form blots. 

Anti-connexin 43: The blots were blocked for 30 min with 2% casein in PBS- 0,1% Tween 20 and incubated for 2 h at room temperature with rabbit polyclonal antibody against mouse monoclonal anti-Cx43 antibody (1/13000; Sigma Aldrich, St. Louis, MO, USA). Blots were then incubated for 1 h at room temperature with anti-mouse immunoglobulin G (IgG) (Fab specific)-horseradish peroxidase (HRP) (1:2500; Sigma Aldrich, St. Louis, MO, USA). 

Anti-phospho-connexin 43: The protein blots were blocked in 5% BSA in Tris-buffered saline Tween 20 (TBST) for 1h and then incubated with phospho-Cx43/GJA1 (Ser368) rabbit polyclonal antibody (1:1000 in TBST; ABCAM, UK) at 4 °C overnight, followed by incubation with HRP-conjugated goat anti-rabbit secondary antibody (1:10000; Bethyl, Montgomery, TX, USA) in 5% non-fat milk-TBST at room temperature for 1h. For loading control, blots were further stripped and re-probed with GAPDH mouse monoclonal antibody (1:10000; Proteintech, Rosemont, IL, USA) followed by HRP-conjugated goat anti-mouse secondary antibody (1:2500; Sigma Aldrich, St. Louis, MO, USA). Bands were visualized using the colorimetric system 3,3’,5,5’-tetramethylbenzidine (TMB) one-component HRP membrane substrate (SurModics, Eden Praire, MN, USA) and quantified by densitometry using ImageJ software (National Institute of Health, NIH, Bethesda, MD, USA).

### 2.10. Immunohistochemistry

Mice were anaesthetized using i.p. injection of ketamine (100 mg/kg) and xylazine (16 mg/kg) and then perfused intracardially with ice-cold 0.1M PBS pH 7.4 and then with ice-cold freshly prepared solution of 4% paraformaldehyde (PFA) in 0.1M PB. The brains were immediately removed, post-fixed for 1 h in 4% PFA at room temperature, rinsed overnight in PBS and then cryoprotected for 24–48 h in 30% sucrose at 4 °C. The brains were frozen in isopentane (‒40 °C) and coronal sections (40 µm thick) of the hypothalamus or brainstem were made with a cryostat (Leica CM3050, France) and collected serially in 0.1M PBS.

For Glial Fibrillary Acidic Protein (GAFP), Ionized Ca^2+^ Binding Adapter Molecule 1 (IBA1) and Cx43 immunostainings, sections were incubated for 1 h saturation buffer containing 0.3% triton X-100 in 1X PBS buffer and respectively 3% normal goat serum (NGS for GFAP), 3% horse serum (HS for IBA1) and 3% NGS coupled with 1% bovine serum albumin (NGS + BSA for Cx43). Primary antibodies against GFAP were used at 1/1000 (mouse #G3893, Sigma Aldrich, St. Louis, MO, USA), against IBA1 at 1/500 (rabbit #019-19741, Wako, Richmond, VA, USA) and against Cx43 at 1/3000 (rabbit #ACC-201, Alomone Labs, Jerusalem, Israel) or 1/2000 (mouse #C8093, Sigma Aldrich, St. Louis, MO, USA). After one night of incubation at 4 °C, labelings were revealed with corresponding secondary antibody coupled with Alexa Fluor 488 or 594 (Invitrogen, Carlsbad, CA, USA), used at 1/400 during 2 h incubation at room temperature.

Detection of biotinylated TAT-GAP19 and TAT-GAP19-I14A was performed after 1 h saturation of slices in 3% NGS and 0.3% triton PBS buffer and 2 h of incubation with avidine FITC (Vector) in the same PBS buffer.

For c-Fos immunohistochemistry, sections were incubated for 10 min in 0.1 M PBS containing 1.5% H_2_O_2_ for quenching of endogenous peroxidase activity. After 1 h in saturation PBS buffer containing 3% NGS and 0.3% triton X-100, sections were incubated for 48 h at 4 °C with a rabbit anti-c-Fos antibody (1/10 000, #PC38, Calbiochem). A biotinylated goat anti-rabbit IgG (1/400, #BA1000 Vector) was used as a secondary antibody (incubated for 1 h 30 at room temperature). Peroxidase activity was revealed using the avidin-biotin complex (1/200, Vector Labs, Burlingame, CA, USA) and diaminobenzidine as chromogen. Non-specific labelling was observed on adjacent sections that were treated identically but without the primary antibody. The reaction was closely monitored and terminated by washing the sections in distilled water when optimum intensity was obtained (3–5 min).

After rinse in PBS, slices were mounted on gelatin-coated slides and coverslipped with Mowiol mounting medium. 

### 2.11. Image Acquisition and Data Processing

Fluorescence images were acquired by confocal microscopy (Zeiss LSM 710). In double-labeling experiments, images were sequentially acquired. All images were further processed in Adobe Photoshop 6.0, with only contrast and brightness adjustments. Photomicrographs labeling were taken on a Nikon Eclipse E600 light microscope using a DXM 1200 Camera equipped with ACT-1 software. The microscope was set at a specific illumination level, as was the camera exposure time. c-Fos-positive nuclei were counted using NIH ImageJ software. Labeling and counting were performed on every fourth coronal section throughout the rostro ¬¬–¬caudal axis from the brainstem to the hypothalamus.

### 2.12. Stereological Estimation of Cx43/Bassoon Apposition

Immunohistochemistry for 1st order stereologic quantifications: The PFA perfusion, cryoprotection and cryosection were performed as previously described [3]. Five animals (15 weeks C57Bl6 mice) per condition at least (ad libitum food and water) were used to perform these quantifications. The DVC and the arcuate nucleus (ARC) in the hypothalamus were cut in 40 µm sections and immunohistochemistry was performed on free-floating sections [3]; saturation solution for 1 h at room temperature, 2% BSA and 0.3% Triton X-100, then incubations using mouse primary antibody against Bassoon (#SAP7F407 Enzo Life Sciences, Farmingdale, NY, USA) at 1/3000 during 3 days at 4 °C, then secondary antibody (goat) directed against mouse IgG and coupled with Alexa 488 (1/400, overnight, 4 °C). Once again, the same saturation solution was used before the Cx43 rabbit primary antibody (#ACC-201 Alomone Labs 1/3000, Jerusalem, Israel) and goat anti-rabbit-IgG secondary antibody (1/400) on the same conditions. The long incubation times are necessary to obtain a strong and constant fluorescent signal throughout the entire section. The sections were serially mounted on slides, using Fluoroshield mounting medium (#104136, Abcam, Cambridge, UK) and then coverslipped. 

Cavalieri method: the tissue was serially cut with uniform thickness of 40 µm. All the sections were taken into account. An entire DVC theoretically comprises 32 sections from rostral to caudal axis. NTS was present on all slices. In the hypothalamus, only the ARC including the median eminence (ME) is considered. It represents a mean number of 12 sections to match the entire nucleus. The total volumes of NTS and ARC were estimated by manually delimiting areas on section images. The reliability of these reference volume estimations was assessed by calculating a coefficient of error (CE) for each NTS and ARC of each mouse as described by Gundersen and Jensen [18]. The estimation of the reference volume was considered accurate if the CE is lower than 7%.

Optical fractionator method: Depending on the average volume of the DVC, an x =2 00 µm and y = 300 µm grid was created to capture circa 150 voxels per animal. This grid was adapted for the ARC (200 µm × 200 µm). Confocal mosaic images were taken and this grid was randomly applied on each mosaic and every intersection within the considered structures was recorded as a voxel position. Voxels of 64 µm^3^ (4 × 4 × 4 µm, z stacks=13) included a 27 µm^3^ zone (3 × 3 × 3 µm) in the center of the voxels. Bassoon fluorescent signals were counted inside this zone. In order to restrict any overestimation, the points limiting the lower and right edges, the corners except the upper left one, the 2 first z-stacks and 2 last z-stacks (over 14) were excluded. The highlighted Bassoon fluorescent signal population was then split into 2: one with every Bassoon signal closes enough (less than 0.5 µm in every dimension of pure black) from a Cx43 signal and the other population with the remaining Bassoon signals. This counting includes the Cx43 signal out of the 27 µm^3^ zone. We considered a point more than 0.1 µm-diameter as a significant fluorescent signal labeling of proteins. To avoid overestimation, two points should not have had any fluorescent connections in the 3 dimensions and must have had at least one z-stack of no-signal black between them. For each structure, a shrinkage coefficient was applied on voxels’ data according to the confocal measured thicknesses of sections. Note that for rostral levels of DVC, where the NTS is in touch with fourth ventricle, another coefficient reported the shrinkage differences between the voxels in the voxels’ thickness depending on their distances from ventricle. These coefficients were calculated after establishing the average thickness profiles of DVC or ARC sections for each animal and then averaging for all. The estimation of the total number of Bassoon signal close to Cx43 signal described previously was carried out according to West and colleagues [19]. 

### 2.13. Ethidium Bromide Uptake and Quantification

Ethidium Bromide Uptake was performed on acute hypothalamic slices prepared from 7-week-old mice fed ad libitum. Mice were euthanized by cervical dislocation followed by decapitation. Brains were removed and placed in ice-cold oxygenated (95% O_2_; 5% CO_2_) sucrose-enriched artificial cerebrospinal fluid containing (aCSF; in mM): 87 NaCl; 75 sucrose; 25 NaHCO_3_; 10 Glucose; 7 MgCl_2_, 2.5 KCl, 1 Na_2_HPO_4_; 0.5 CaCl_2_; 320–330 mOsm. The lateral borders of the brain were removed and coronal slices (300 µM thick) containing the mediobasal hypothalami were cut with a vibratome (VT2000S; Leica, Wetzlar, Germany). Slices were transferred to a constantly oxygenated (95% O_2_; 5% CO_2_) holding chamber containing sucrose-enriched aCSF for 30 min, then at least 30 min in the recording aCSF used for dye uptake assay containing: 119 NaCl; 26.2 NaHCO_3_; 11 Glucose; 2.5KCl; 2.5 CaCl_2_; 1.3 MgCl_2_; 1 Na_2_HPO_4_; 320–330 mOsm. Then, the slices were then randomly distributed in 10 mL submerged chambers and incubated for 15 min in recording aCSF, or in aCSF supplemented with carbenoxolone 200 µM or TAT-GAP19 100 µM before and during the application of EtBr (314 Da, 4 µM, 10 min). Slices were then washed for 20 min in aCSF and fixed for 1 h 30 in PBS containing 4% PFA before immunohistochemistry. Slices were treated with PBS containing 3% NGS, 1% Triton X-100 to block nonspecific binding sites. Slices were then incubated overnight with rabbit polyclonal anti-GFAP (1/1000, #Z03334, Dako, Santa Clara, CA, USA), washed in PBS and incubated for 1 h 30 with secondary antibody goat α-Rabbit-Alexa Fluor 488 (1/400, #A-11034, Invitrogen, Carlsbad, CA, USA). After rinse in PBS, slices were incubated 1 min with Hoechst 1/10000, rinsed, and mounted on gelatin-coated slides and coverslipped with Mowiol mounting medium. Images were acquired on a confocal microscope (ZEISS LSM 710) using the 488-nm (Alexa), 561-nm (BET) and 405 nm band. Dye uptake was evaluated in GFAP labelled cells and expressed as the difference between the fluorescence measured in GFAP+ astrocyte nuclei and the background fluorescence measured where no nuclei were detected. Values of fluorescence in different experimental conditions were normalized relative to the control level.

### 2.14. Statistical Analysis 

All results are presented as mean ± SEM. Statistical analysis was performed using GraphPad Prism 6. Comparison between two groups was performed using unpaired two-tailed student’s t-test. One-way ANOVA was performed for comparisons between mice treated with different conditions followed by Dunnett’s post-test. A repeated-measure ANOVA was performed when needed.

## 3. Results

### 3.1. Glial Cx43 Expression in the Hypothalamus and DVC

To characterize connexins (Cxs) expression within the hypothalamus and brainstem, a quantification of mRNA coding for Cx43, 30, 26 and Pannexin1 (Panx1) genes were performed by reverse transcriptase quantitative polymerase chain reaction (RT-qPCR). Panx1 is a protein that also forms transmembrane hexameric HCs. The relative proportion of each of these mRNAs was evaluated in the hypothalamus (*n* = 23), the DVC (*n* = 24), and compared with the cortex (*n* = 6) of control animals (Figure 1A).

In all structures studied, Cx43 mRNAs were the most abundantly expressed transcripts when compared to Cx30, Cx26 and Panx1 (Figure 1A). Moreover, Cx43 mRNA expression in the hypothalamus and DVC was significantly higher than the one in the cortex (hypothalamus: + 47.4% *p* < 0.01; NTS: + 45.9% *p* < 0.01; Figure 1A). 

Immunohistochemistry confirmed the robust Cx43 expression within the hypothalamus and the DVC (Figure 1B,C). At the hypothalamic level, Cx43 immunostaining revealed a maximum density at the ARC level (Figure 1B) and in the wall of the third ventricle (Figure 1C). In the DVC, NTS has a high density of Cx43 signal, differentiating it from other bulbar structures (Figure 1D). It is noteworthy that Cx43 is weakly present in the DMNX and almost absent from the area postrema (AP, Figure 1D). Moreover, as expected, Cx43 staining was found strongly associated with GFAP+ protoplasmic astrocytes from both structures (Figure 1B–I). A double-staining of Cx43 with vimentin revealed the association of Cx43 with hypothalamic tanycytes (Figure 1F) and tanycytes-like cells i.e., vagliocytes [20] of the DVC (not shown). Finally, co-staining of Cx43 with Ionized Ca^2+^ Binding Adapter Molecule 1 (IBA1), a marker of microglia, showed that Cx43 was sometimes found associated with these cells in the DVD Figure 1K–J and hypothalamus (not shown).

### 3.2. Modulation of Cx43 Expression by Nutritional Status

The variations in Cx43 and GFAP transcript expressions were next evaluated by RT-qPCR within the hypothalamus and DVC after 24 h of fasting and refeeding (2 and 4 h). In the hypothalamus, the 24 h of fasting decreased significantly the expression of mRNAs coding for Cx43 (−13% *p* < 0.001; Figure 2A).

This decrease was not reversed by short-term refeeding (2 h: −18%; 4 h: −26% *p* < 0.001; Figure 2A). Interestingly, 24 h of fasting also reduced GFAP mRNAs expression (−18% *p* < 0.05; Figure 1A), a decrease that was partly reversed 4 h after refeeding (Figure 2A). In the DVC, no variation of Cx43 and GFAP transcripts expression was observed after fasting or refeeding (Figure 2B). The reduced hypothalamic expression of Cx43 induced by 24 h fasting was confirmed by Western blot (Figure 2C). Moreover, the level of Cx43 phosphorylation on serine 368 (pS368) was also significantly reduced by fasting (Figure 2C) suggesting a possible post-transcriptional modulation of Cx43 activity in addition to the reduced expression. In high fat diet (HFD)-fed mice, we observed an upregulation of the transcripts coding for Cx43 and GFAP within the hypothalamus (+54%, *p* < 0.001 and +85%, *p* < 0.001 for Cx43 and GFAP respectively; Figure 2D) while no significant difference was observed within the DVC (Figure 2E). Moreover, in the hypothalamus, the level of Cx43 phosphorylation quantified on S368 was also significantly increased by HFD (+45%, *p* < 0.001; Figure 2F).

### 3.3. Perisynaptic Localization of Cx43 

We next sought to determine whether Cx43 positive structures (GJs and HCs) are positioned perisynaptically within the hypothalamus and DVC and thus potentially able to release bioactive molecules and to modulate neuronal networks involved in energy balance control. To this goal, we performed a double Cx43 and bassoon labeling. The cytomatrix protein bassoon is a well-established marker of presynaptic structures [21]. Low magnification confocal microscopy at the hypothalamus (Figure 3A) and brainstem (not shown) level showed a Cx43 labeling consisting in dispersed puncta throughout the tissue (Figure 3A).

Bassoon immunoreactivity consisted of similarly dispersed puncta, as well as short bars of puncta (Figure 3A,B). In high magnification confocal microscopy, it appears clearly that many Cx43 puncta are located near presynaptic elements (Figure 3B), going in some cases up to the juxtaposition (Arrows in Figure 3B). We next sought to estimate the proportion of Cx43/bassoon juxtaposition within the ARC and NTS using a first-order stereologic quantification approach (Figure 3C–E). The total volumes of NTS and ARC were estimated from manually delimited areas on section images (Figure 3E). A grid was then created to capture circa 150 voxels per animal and structures. Each voxels of 64 µm^3^ (4 µm sized cube, z stacks = 13; Figure 3C) included a 27 µm^3^ zone (3 µm sized cube) in the center of the voxels where bassoon fluorescent signals have been counted. The percentage of bassoon fluorescent signal close enough from a Cx43 signal (less than 0.5 µm in every dimension) was thus quantified. This stereological estimation of Cx43/bassoon appositions showed that a high proportion of bassoon elements were found in juxtaposition to Cx43 signals in both the NTS (85.13%) and ARC (80.46%) nuclei (Figure 3D).

### 3.4. Cx43 HCs Activity in Hypothalamic Slices 

Next, we investigated the activity of Cx43 HCs by performing the EtBr uptake assay in hypothalamic slices. EtBr uptake is s a functional index of hemichannel activity including Cx43 HCs [22]. Slices were prepared from mice fed ad libitum. In control conditions (aCSF medium), fluorescence measurements showed that vehicle-treated hypothalamic slices took up a large amount of EtBr over a 10 min period (Figure 4A), suggesting a high basal HCs activity in the ARC.

As expected, a significant proportion of EtBr labeling was found in GFAP+ astrocytes (Figure 4A,B). Carbenoxolone (Cbx, 200 µM) was used as a positive control for hemichannel blockade. Hypothalamic slices treated with Cbx displayed a strong reduction in EtBr uptake compared with vehicle control (Figure 4A,B). We next tested the effect of TAT-GAP19 on EtBr uptake; TAT-GAP19 is designed to specifically inhibit Cx43 HCs. TAT-GAP19 100 µM significantly reduced EtBr uptake from the ARC when compared to untreated condition, albeit this inhibition was slightly less broad than observed with Cbx (Figure 4A,B). This TAT-GAP19 100 µM dose was chosen based on the previous work of Abudara and colleagues [12]. Importantly, this inhibition of EtBr uptake induced by TAT-GAP19 was significant in GFAP+ positive astrocytes (−59% and −52% in Cbx and TAT-GAP19 treated slices respectively; Figure 4B). 

### 3.5. TAT-GAP19 Diffusion after i.c.v Injection

In order to inhibit Cx43 HCs activity in vivo, particularly in both the hypothalamus and brainstem, we choose to perform i.c.v. (lateral ventricles) injection of TAT-GAP19 on chronically cannulated mice. We first checked for the diffusion of TAT-GAP19 into brain parenchyma after its i.c.v. administration. To this goal, we carried out staining of biotinylated TAT-GAP19 by avidine-FITC. One hour after i.c.v. injection of the peptide 25 µg, we observed that biotinylated TAT-GAP19 was strongly present in the parenchyma lining the lateral ventricle near the injection site (Figure 5A,B).

In addition, biotinylated TAT-GAP19 peptide was found only located in structures surrounding the third ventricle and mainly in hypothalamic nuclei (Figure 5B). More caudally, a labelling was observed in structures bordering the fourth ventricle (Figure 5C) including the AP and NTS (Figure 5D). It should be noted that no significant labelling could be visualized in the other central structures suggesting that when injected in lateral ventricles TAT-GAP19 targets quite specifically areas surrounding ventricles. Biotinylated TAT-GAP19 peptide could be detected within the hypothalamus parenchyma as soon as 30 min after its i.c.v. injection (Figure 5D) and diffusion area increased with time to cover up the hypothalamus after 2 h (Figure 5E–G). Importantly, the diffusion of the TAT-GAP19-I14A was strictly identical to those of native peptide (Figure 5H). 

### 3.6. i.c.v. TAT-GAP19 Injection Modifies Food Intake and Meals Microstructure

We next tested whether a blockade of the HCs activity could modify food intake in mice. To this goal, we performed i.c.v. injections (lateral ventricles) of TAT-GAP19 (12.5 and 25 µg) at the beginning of the dark phase on chronically cannulated mice. A central injection of TAT-GAP19 12.5 µg induced a modest and transient decrease in food intake observable only during the 3 h following the injection (Figure 6A).

At the dose of 25 µg, TAT-GAP19 significantly reduced intake food as soon as 3 h and during the 24 h post-injection (Figure 6A). Over the first 12-h post-injection period, this decrease reaches 41% and persisted until 24 h (*p* < 0.001; Figure 6A). Importantly, TAT-GAP19-I14A, which does not recognize Cx43, did not modify food intake when injected at 25 µg (Figure 6A). To decipher feeding behavior investigation during TAT-GAP19 treatment, meals microstructure analysis (Figure 6B) was then computed for dark phase period i.e., 0–12 h. Compared with the vehicle, TAT-GAP19 reduced meal number by 56.8% during the 0–12 h period (16.7 +/− 1.3 min vs. 7.2 +/− 1.1, *p* < 0.05; Figure 6C) and diminished also meal size by 44.6% (153.4 +/− 22.4 mg vs. 85.9 +/− 15.5 mg, *p* < 0.01; Figure 6D) without modifying meal duration (Figure 6E). Finally, intermeal interval was increased by 2.8 fold by TAT-GAP19 treatment (Figure 6F). The impact of TAT-GAP19 (12.5–50 µg) was also evaluated on energy expenditure and on locomotor activity (activity spontaneous). The results obtained show that TAT-GAP19 did not modify energy expenditure during the dark phase. Figure 6G illustrates the absence of effect of TAT-GAP19 on energy expenditure with the higher dose (50 µg). Importantly, locomotor activity was not affected by TAT-GAP19 injection during the period where food intake was diminished by the treatment (Figure 6H,I). 

### 3.7. Effect of TAT-GAP19 on Glycemic Regulation during a Carbohydrate Overload

Since we observed a strong impact of i.c.v. TAT-GAP19 injection on food intake, we next evaluated the impact of this central TAT-GAP19 administration on glycaemia and regulatory mechanisms operating during a hyperglycaemia (Figure 7).

In fasted animals, TAT-GAP19 administration (12.5–50 µg) did not modify glycaemia (Figure 7A,B). Moreover, we performed an oral glucose tolerance test (OGTT) 30 min after the i.c.v. injection of TAT-GAP19 (12.5–50 µg). The absence of effect of TAT-GAP19 on glycaemia parameters is illustrated for the higher dose (50 µg). As observed before, TAT-GAP19 did not significantly change blood sugar within 30 min that follow its administration (Figure 7C). As expected in control animals, *per os* administration of glucose induced a rapid increase in blood sugar. After 10 min, the rate of blood glucose gradually returns to its normal value under the effect of insulinemic regulation. In mice having received a central injection of TAT-GAP19, blood glucose levels were comparable to those obtained in the control individuals (Figure 7C,D).

### 3.8. c-Fos Expression in Response to i.c.v. Administration of TAT-GAP19

Central structures activated in response to i.c.v. administration of TAT-GAP19 25 µg were identified using the immune detection of the early gene c-Fos performed on animals sacrificed three hours after treatment. A very low basal level of c-Fos positive nuclei was observed in the brainstem, pons and forebrain of NaCl-treated mice (Figure 8).

TAT-GAP19-treated mice exhibited a strong rise in the number of c-Fos positive nuclei in a very limited number of brain structures including bulbar and hypothalamic nuclei i.e., NTS, PVN and ARC (Figure 8). Interestingly, this pattern of TAT-GAP19 -induced cellular activation was restricted to centers involved in the control of food intake. In addition, a significant c-Fos labelling was also observed in the wall of third and fourth ventricles of TAT-GAP19-treated mice (Figure 8A–O).

## 4. Discussion

The present work shows that the two main structures involved in homeostatic regulation of energy balance are characterized by a notable expression of Cx43 associated with high glial density. Interestingly in the hypothalamus Cx43 expression is sensitive to nutritional status changes (fasting, obesity). In addition, a strong basal HCs activity in the ARC is suggested by the EtBr uptake assay. This HC activity is at least partially dependent on Cx43. Furthermore, the central administration of TAT-GAP19, a permeant peptide, that inhibits specifically the opening of Cx43 HCs, induces a decrease in food intake, without affecting locomotor activity. Finally, i.c.v. TAT-GAP19 administration induces cellular activation attested by c-Fos expression specifically in the hypothalamus and the DVC. Altogether, these results constitute the first demonstration that glial Cx43 HCs are involved in the regulation of food intake.

### 4.1. Cx43 Expression within the Hypothalamus and DVC and Modulation by Nutritional Status

Cx43 is a ubiquitous protein found in almost all tissues of the body except red cells, sperm and skeletal muscle cells [7]. Within the CNS, this protein is expressed by glia, mainly astrocytes and to a lesser extent microglia [9]. Exceptionally, neuronal Cx43 expression has been reported in olfactory bulb neurons [23] and spinal cord motor neurons [24], where Cx43 is involved in the formation of electrical synapses. We first investigated Cx43 expression in the hypothalamus and DVC, two major centers liable for the regulation of the energy balance including homeostatic control of food intake. An analysis of transcripts encoding proteins known for their ability to form GJs and/or HCs in glial cells (Cx43, Cx30, Cx26 and Panx1) has shown that Cx43 is the most widely expressed in the hypothalamus and DVC. These structures are enriched in Cx43 mRNA when compared to the cortex, a structure taken here as a reference and reported to express high level of Cx43 in the adulthood [25]. This strong Cx43 expression in the hypothalamus and DVC has been confirmed at the protein level by immunohistochemistry. These two structures are distinguished by a higher Cx43 immunoreactivity than surrounding structures. At the brainstem level, the Cx43 labelling clearly delimits the contours of the NTS. This high density Cx43 expression was found to a lesser extent at the hypothalamic level around the third ventricle and in the ARC. At higher magnifications, we confirmed the presence of Cx43 in membranes of GFAP + astrocytes. As previously described [26,27], tanycytes and more precisely β-tanycytes expressed Cx43. Interestingly, a population of tanycytes-like cells located within the DVC and originally described by Pecchi and colleagues [20] also expressed Cx43. Finally, it should also be noted that some Cx43 labeling is also associated with IBA1^+^ microglial cells. This cellular localization reported for the first time in the hypothalamus and DVC is consistent with the results of the literature obtained in other structures from the CNS [9]. 

We then investigated whether Cx43 expression (mRNAs and protein) can be modified by drastic changes in energy balance. Control of Cx expression occurs mainly at the transcriptional level [7] and the short Cx half-life, of the order of only a few hours (1.5 to 4 h for the Cx43, [28], implies a permanent synthesis. We first chose to test a 24 h fast, known to cause a change in the energy substrates used. This change in metabolism is accompanied by a decrease in insulinemia and leptinemia and an increased secretion of ghrelin [29]. Under these conditions, we observed a decrease in Cx43 expression in the hypothalamus. No change in Cx43 expression was observed in the DVC. The important contribution of the hypothalamus to counter-regulatory mechanisms observed during a fast [30] may explain this result. Refeeding (2 and 4 h) did not restore the initial levels of Cx43. Secondly, we looked for variations in Cx43 expression in diet-induced obese (DIO) animals. We observed an upregulation of Cx43 expression in the hypothalamus of DIO animals, while again no modification was observed within the DVC. The turnover, internalization and degradation of Cxs including Cx43 are highly associated with phosphorylation/dephosphorylation events and can be triggered by a variety of stimuli (e.g., growth factors, extracellular matrix interactions, ischemia, inflammation, see [28] for review). Cx43 phosphorylation by protein kinases especially at serine 368 has been associated to i/ GJ internalization, ii/ changes in GJ intercellular communication and iii/ reduced Cx43 half-life [31]. Interestingly, we observed a marked decrease in phosphorylated Cx43 levels in the hypothalamus during fasting while during induced obesity, the level of phosphorylated Cx43 increased.

Using EtBr uptake as an index of HCs activity, we observed a high HCs permeability within the ARC of ad libitum-fed mice. This HCs activity relied significantly on Cx43 HCs, especially in GFAP+ astrocytes, since it was inhibited by TAT-GAP19, a mimetic peptide preventing interaction between the C-terminal part of Cx43 with the L2 loop and thus causing the specific closure of the Cx43 HCs [12]. It is noteworthy that TAT-GAP19 has no influence on communication via the GJ made up of Cx43 [12]. HCs are known to mediate the transfer of signaling molecules i.e., ATP and glutamate, between the cytoplasm and the extracellular space, constituting a non-vesicular route for gliotransmitters release (see [32] for review). In agreement with this concept, blockade of Cx43 HCs with GAP26 resulted in reduced frequency of excitatory post-synaptic currents (fEPSPs) recorded in hippocampal CA1 neurons after Schaffer collateral stimulation via ATP signaling [33]. Furthermore, it was demonstrated that glial Cx43 modulates glutamatergic synaptic activity of CA1 pyramidal cells via changes in synaptically released glutamate [34]. 

In the hypothalamus and the DVC, Cx43 labelling was often found in close proximity, even juxtaposed, to bassoon positive elements. Bassoon is a very large scaffolding proteins of the cytomatrix assembled at the active zone where neurotransmitters are released [21]. Our stereological study revealed that a high proportion of Cx43 signal (~80–85%) was found in the close environment of the synaptic elements (<1 µm). This result is quite different from a pioneer ultrastructural work by Sipe and Moore [35] performed at the lateral hypothalamic area level reporting that about 90% of astrocytic GJ, identified on the basis of morphological criteria, was positioned within a distance of 15–20 µm from a synaptic terminal. On the other hand, our observation is to be compared to the work carried out at the barrel cortex where the average distance between an excitatory synapse and an astroglial Cx43 GJ, identified on electron-microscopic images, was about 0.7 µm [36]. Nevertheless, contrary to cited works, our stereological analysis is not based on the morphological identification of GJ but on Cx43 immunofluorescence staining, which allows the detection of connexons involved in both GJ and HCs. Thus, we can assume that a proportion of Cx43 that we detected in the immediate vicinity of the synapse is not engaged in GJ but could constitute HCs. The high HCs activity we observed in hypothalamus slices support this assumption. 

Altogether, these results suggest that, in structures involved in homeostatic regulation of food intake, Cx43 expression and Cx43 GJ and/or HCs gating can be regulated by hormonal and/or metabolic signals associated with nutritional status. In turn, these modulations could result in a disturbance of astrocytic and tanycytic GJ networks and/or the release of neuroactive compounds through Cx43 HCs. 

### 4.2. The Inhibition of Cx43 HCs Diminishes Food Intake

All of the results discussed above prompted us to test the effect of Cx43 HCs inhibition on food intake of ad libitum-fed mice. We postulated that Cx43 HCs activity could allow the release of neuroactive compounds and hence constitute a tonic modulation of neuronal networks dedicated to the control of food intake. To specifically target Cx43 HCs, we used GAP19 as previously done with EtBr uptake assays. Nonetheless, to facilitate GAP19 diffusion and cell penetration in vivo, we coupled GAP19 to a TAT sequence, derived from HIV. The TAT sequence facilitates cellular internalization of the peptide by a macropinocytosis mechanism [37]. Moreover, we have chosen to administer this peptide by i.c.v. injection in lateral ventricles in order to reach both the hypothalamus and DVC, which border the third and fourth ventricles, respectively. This route of administration, which is not totally specific for the targeted structures, can potentially induce diffusion of the peptide in other brain structures and lead to unspecific behavioral effects. However, based on our previous results [38], we postulated that this would not be the case because of the facilitated peptide diffusion from the ventricles to the ARC and NTS thanks to the proximity of the organs circumventricular, namely the median eminence and the area postrema. FITC labeling of biotinylated TAT-GAP19 or TAT-GAP19-I14A performed at different time points after their i.c.v. administration confirmed a privileged presence of the peptide in the ARC and NTS after its ventricular injection. Thus, we next evaluated the impact of i.c.v. TAT-GAP19 administration performed at the beginning of the mouse activity period. TAT-GAP19 rapidly (<1.5 h) diminished food intake, and this effect persisted throughout the dark phase. A meal microstructure analysis revealed that TAT-GAP19 causes both meal frequency and size decrease. While meal frequency reduction is evocative of a sickness behavior [39], the decrease in meal size refereed to a more physiological process since endogenous substances regulating food intake, i.e., cholecystokinin, leptin, are known to modify the meal size. The possible nonspecific action of TAT-GAP19 inducing a broad distress resulting in anorexia seems unlikely since we did not observe signs of suffering. In accordance, i.c.v. TAT-GAP injection did not modify locomotor activity. In addition to food intake, we looked for a possible effect of TAT-GAP19 on carbohydrate regulation. The permeability of these HCs for the glucose suggests a role for Cx43 in central glucodetection, and it was previously reported that the inhibition of Cx43 synthesis, by interfering RNAs (siRNAs) injected into the rats hypothalamus parenchyma, decreases hypothalamic glucose sensitivity and glucose sensing-induced insulin secretion [40]. Here, the TAT-GAP19 peptide did not alter glycaemia or insulinemic regulation following a glucose overload. The differences in the protocols used, i.e., a reduction of Cx43 expression by siRNAs and acute inhibition of Cx43 HCs activity could explain this apparent discrepancy. Finally, mapping of cellular activation using c-Fos immunolabelling [41] was carried out on sections of the bulbar region and of the entire diencephalon. At a time where anorexia is ongoing, the i.c.v. TAT-GAP19 peptide injection induced cellular activation only in the hypothalamus and DVC. This activation profile is characteristic of cellular activation obtained after i.c.v. administration of appetite suppressant compounds [3,5,17,39,42]. This result shows that inhibition of Cx43 HCs within the hypothalamus and DVC modifies the activity of circuitries potentially involved in central appetite control, which could explain the diminution of meal number and size observed in the behavioral study. 

### 4.3. Possible Mechanisms Underlying the Reduction of Food Intake Induced by Cx43 HCs Inhibition 

The release of gliotransmitters and their involvement in brain functions are illustrated by numerous results obtained on different physiological and pathological models (see for reviews [43,44]). In this context, the release of neuroactive substances by Cx HCs has been largely proposed (see for review [11]). Astrocytic release of ATP through the Cx26 HCs contributes to the control of breathing [45]. Stehberg and colleagues [13] have shown that the release of neuroactive substances via the astrocytic Cx43 HCs is necessary for long-term memorization of fear conditioning. Indeed, injection of TAT-Cx43L2 or GAP27 into the basolateral amygdala resulted in amnesia to fear conditioning training. Interestingly, co-administration of these blocking peptides with a gliotransmitter cocktail including glutamate, glutamine, lactate, D-serine, glycine, and ATP restored fear conditioning memory. Nevertheless, this study did not identify a specific gliotransmitter in mediating the effects of HCs on learning and memory. Similarly, the nature of neuroactive substance(s), released by hypothalamic and brainstem glial cells through Cx43 HCs, capable of modulating food intake remains to be identified. This issue appears as a puzzle since most of the compounds transported through Cx43 HCs have a profound impact on food intake. ATP, amino acids and their metabolites including glutamine glutamate, aspartate, glycine or D-serine can directly or indirectly (via modulation of neurotransmitters release) modify feeding behavior [46]. Interestingly, most of them were known for their orexigenic action. For instance, glutamate was reported to stimulate feeding behavior by acting at the hypothalamic level [47]. ATP appears also as a potential candidate. Its release through Cx43 HCs has been widely reported [11], and this compound exerts orexigenic effects thanks to its action on P2X2 [48] and P2Y [49] receptors. 

Another intriguing question that remains open is that of the glial cells identity at the origin of food intake regulation via HC Cx43. Presumably, GFAP + astrocytes are the primary cell type involved in such regulation given i/ their involvement in the tripartite synapse and astrocyte-neuron communications (see for review [50]) and ii/ the strong Cx43 expression that we observed in GFAP+ cells within the hypothalamus and the DVC. However, tanycytes should not be overlooked. Indeed, we observed that Cx43 is expressed by a subpopulation of tanycytes both within the hypothalamus and DVC. Given their contribution to the regulation of food intake that has emerged in recent years [1,3,51,52], we can assume that tanycytes are susceptible to release, via Cx43 HCs, which are neuroactive substances acting on energy balance regulation. This is often the approach we used, i.e., icv injection of TAT-GAP19, as is it can clearly inhibit Cx43 HCs activity in the cells bordering both the third and fourth ventricles. Finally, it was proposed that microglia cells also contribute to the modulation of energy balance [1]. Albeit, Cx43 seems to be scarcely expressed by hypothalamic and brainstem microglia, we cannot totally rule out the possibility that microglial Cx43 HCs inhibition underlines the effect of TAT-GAP19 on food intake. The deletion of the Cx43 gene in specific cell type could partly answer this question. 

In summary, this study provides the first evidence indicating that inhibition of Cx43 HCs activity interferes with the control of food intake and suggests a possible tonic glial delivery of orexigenic molecules via Cx43 HCs. By reducing food intake, the inhibition of glial Cx43 could constitute a new therapeutic avenue against overweight, obesity and their comorbidities. More broadly, this work helps to support the hypothesis of the involvement of the glial compartment in regulating the energy balance.

## Figures and Tables

**Figure 1 cells-09-02387-f001:**
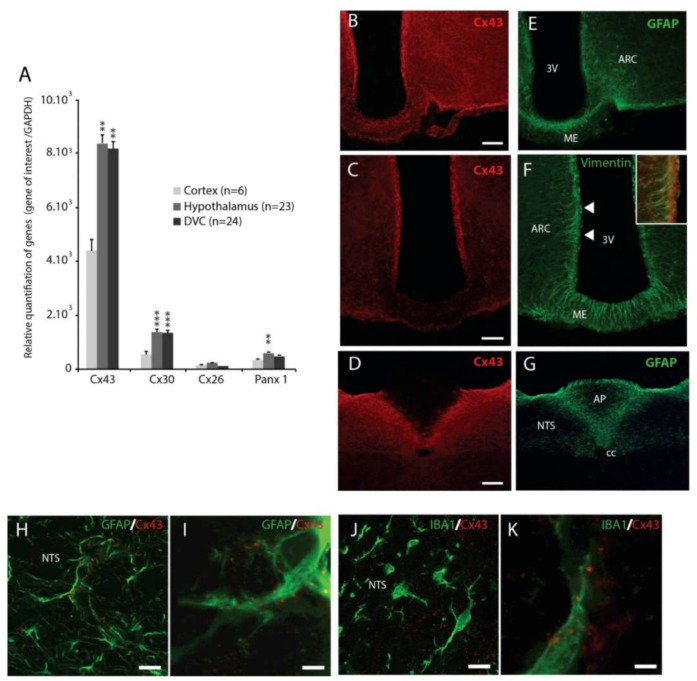
(**A**): Glial connexins and pannexin1 mRNAs quantification measured by reverse transcriptase quantitative polymerase chain reaction (RT-qPCR) in the cortex, hypothalamus and DVC of control mice (*n* = 6, 23 and 24 respectively). ** *p* < 0.01 and *** *p* < 0.001 compared to cortex. (**B**–**G**): Confocal images of Connexin-43 (Cx43) (red) and Glial Fibrillary Acidic Protein (GFAP) (**E**–**G**) or vimentin (**F**) double labeling immunohistochemistry performed within the hypothalamus and Dorsal Vagal Complex (DVC). Scale bar: 200 μm. Inset in F shows Cx43 and vimentin co-localization in α-tanycytes. (**H**,**I**): Confocal images Cx43 (red) and GFAP (green) double immunofluorescence within the DVC. Cx43 is strongly associated with astrocytes GFAP+ (green) processes. (**J**,**K**): Confocal images of Cx43 (red) and Ionized Ca^2+^ Binding Adapter Molecule 1 (IBA1) (green) double immunofluorescence within the DVC. Note that Cx43 labelling is sometimes found associated with thin processes of IBA^+^ microglia. Scale bars: 20 μm for (**H**, **J**) and 2 μm for (**I**) and (**K**). AP: area postrema, NTS: nucleus tractus solitarius, cc: central canal; ARC: arcuate nucleus, 3V: third ventricle, ME: median eminence.

**Figure 2 cells-09-02387-f002:**
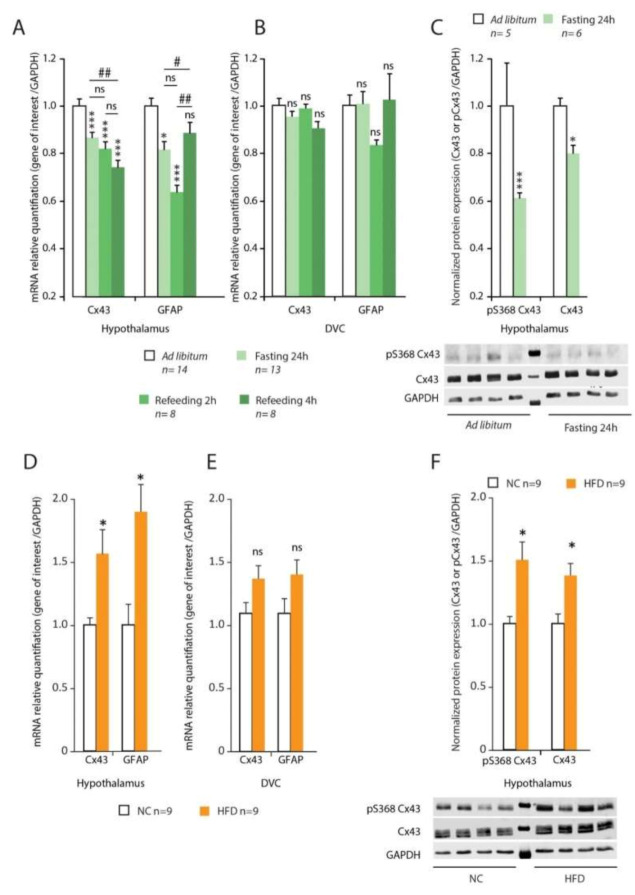
(**A**,**B**): Connexin-43 (Cx43) and Glial Fibrillary Acidic Protein (GFAP) mRNA expression (normalized with Glyceraldehyde-3-phosphate dehydrogenase gene, GAPDH) was assessed by reverse transcriptase quantitative polymerase chain reaction (RT-qPCR) from the hypothalamus (**A**) and dorsal vagal complex (DVC) (**B**) of control ad libitum (*n* = 14), 24 h fasted (*n* = 13), 24 h fasted plus 2 h refeed (*n* = 8) or plus 4 h refeed (*n* = 8) mice. (**C**): Representative Western blot analysis of phosphorylated Cx43 (pS368) and Cx43 expression in the hypothalamus of control ad libitum (*n* = 5) and 24 h fasted (*n* = 5) mice. (**D**,**E**): Cx43 and GFAP mRNA expressions (normalized with GAPDH) were assessed by Reverse Transcription quantitative Polymerase Chain Reaction (RT-qPCR) from the hypothalamus (**D**) and dorsal vagal complex (DVC) (**D**) of Normal Chow (NC) (*n* = 9) and High Fat Diet (HFD) fed (*n* = 9) mice. (**F**): Representative Western blot analysis and quantification of phosphorylated Cx43 (pS368) and Cx43 expression in the hypothalamus of NC (*n* = 9) and HFD-fed (*n* = 9) mice. A two-way analysis of variance test was performed between different experimental groups * *p* < 0.05 and *** *p* < 0.001 significantly different from control ad libitum. #*p* < 0.05 and ##*p* < 0.001 significantly different from 24 h fasted mice. ns: no significant difference.

**Figure 3 cells-09-02387-f003:**
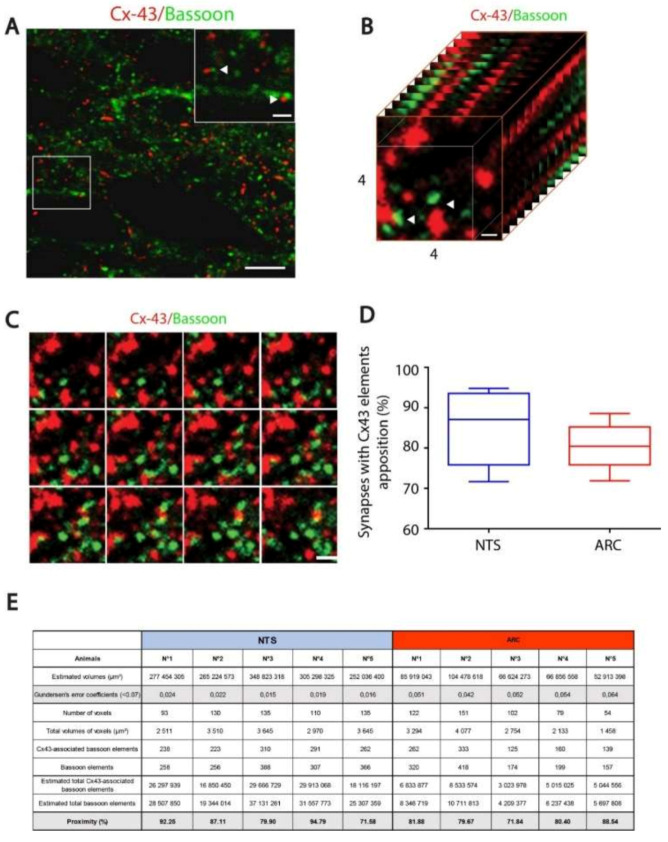
(**A**): Confocal images of connexin-43 (Cx43) (red) and bassoon (green) double labeling immunohistochemistry performed within the nucleus tractus solitarius (NTS). Scale bar: 5 μm. Inset: Arrows show Cx43 labelling in close vicinity of synapses labelled by presynaptic bassoon protein. Scale bar 1 µm. (**B**): An example of a voxel (4 × 4 µm) used for the quantification of Cx43/bassoon appositions. Scale bar: 0.5 µm. (**C**): Twelve z-stack images defining the voxel illustrated in B and allowing the identification of Cx43/bassoon appositions. (**D**): Stereological estimation of Cx43/bassoon appositions in the NTS (85.13%) and arcuate nucleus of the hypothalamus (ARC) (80.46%) nuclei. (**E**): Table gathering the different stereological values acquired or estimated for each studied animal.

**Figure 4 cells-09-02387-f004:**
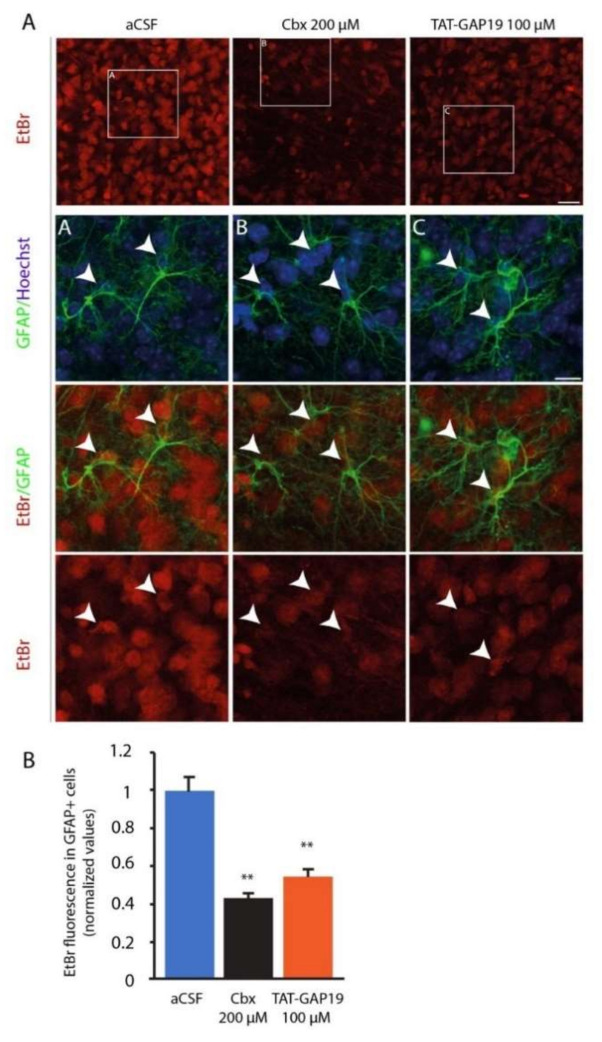
(**A**): Representative images depicting ethidium bromide (EtBr) uptake via hemichannels in hypothalamic (ARC) slices treated with either artificial cerebrospinal fluid (aCSF), 200 μM carbenoxolone (Cbx) or 100 µM the connexin-43 (Cx43) hemichannel blocker TAT-GAP19 for 15 min, then exposed to EtBr for 10 min. After fixation, slices were submitted to Glial Fibrillary Acidic Protein (GFAP) immunostaining and Hoechst labelling. Scale bar: 15 µm. (**B**): Quantification of EtBr fluorescence measured in GFAP+ astrocytes. *n* = 5, with each *n* representing the average of three replicates. ** *p* < 0.01.

**Figure 5 cells-09-02387-f005:**
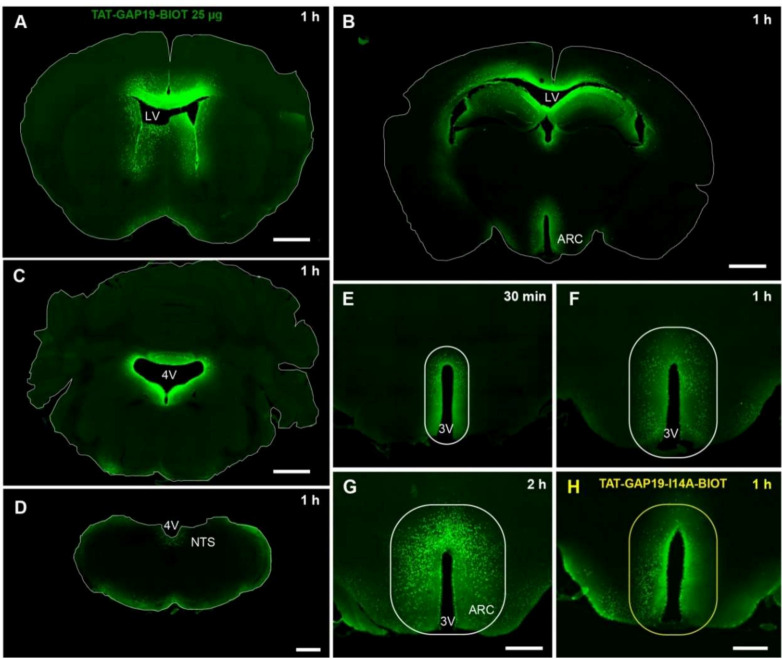
(**A**–**D**): Diffusion of biotinylated TAT-GAP19 into the hypothalamus (**A**,**B**) and hindbrain (**C**,**D**) 1 h after its i.c.v. administration. White lines delimit brain tissue. Scale bars: 1 mm in (**A**) and (**C**); 500 μm in (**C**) and (**D**). (**D**–**F**)**:** High magnification of biotinylated TAT-GAP19 25 µg diffusion within the hypothalamus 30 min (**D**), 1 h (**E**) and 2 h (**F**) after its injection into the lateral ventricles. Scale bar: 500 µm. (**G**): High magnification of biotinylated TAT-GAP19-I14A 25 µg diffusion within the hypothalamus 1 h after its injection into the lateral ventricles. Scale bar: 500 µm. (**H**): High magnification of biotinylated TAT-GAP19-I14A diffusion within the hypothalamus 1 h after its injection into the lateral ventricles. White lines (**D**–**G**) and yellow lines (**H**) delimit diffusion area of biotinylated peptides. ARC: arcuate nucleus; 3V: third ventricle; 4V: fourth ventricle; NTS: nucleus tractus solitarius; LV: lateral ventricle.

**Figure 6 cells-09-02387-f006:**
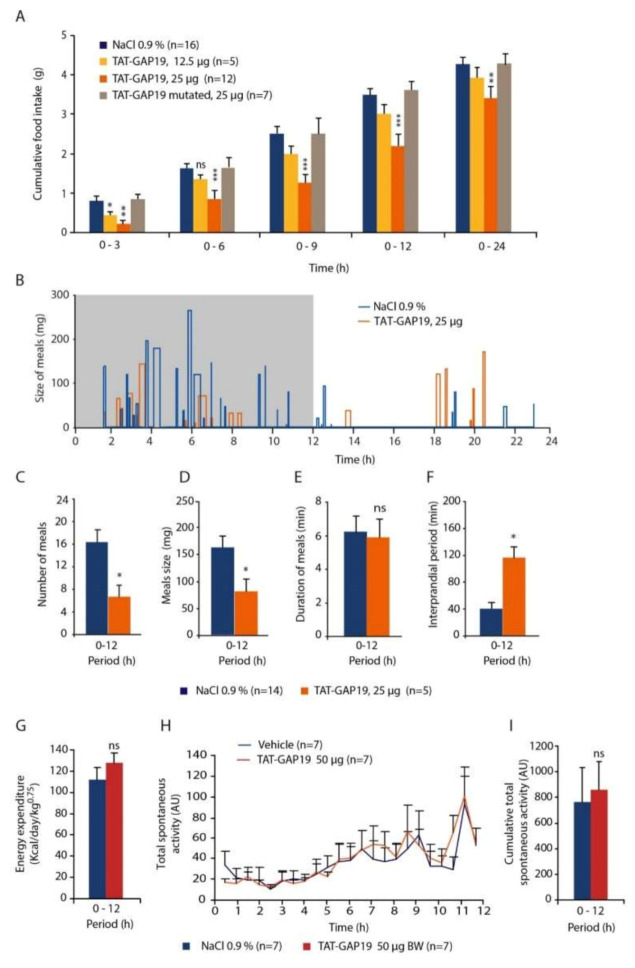
(**A**): Cumulative food intake (g), measured over a 24 h period, of mice having received an i.c.v. injection of either vehicle (Sodium Chloride (NaCl) 0.9%) or TAT-GAP19 (12.5 or 25 µg). The TAT-GAP19-I14A, which includes a point mutation making it ineffective on connexin 43 hemichannels (Cx43 HCs) activity was tested on food intake at the dose of 25 µg. (**B**): Graph showing meal size over 24 h after i.c.v. injection of vehicle or TAT-GAP19 25 µg. Note that each bar represents a meal and its width the meal duration. The dark period is represented by the shaded box. (**C**–**F**): Meal microstructure analysis performed during the dark period for mice that received either vehicle or TAT-GAP19 25 µg allowed quantification of meals number (**C**), meals size (in mg of food, (**D**)), meal duration (in min, (**E**)) and interprandial period (in min, average intervals between meals, (**F**–**G**)): Measure of energy expenditure (in Kcal/day-1/kg^0.75^) within 12 h after injection of 0.9% NaCl vehicle (*n* = 7) or 50 µg TAT-GAP19 (*n* = 7). (**H**,**I**): Measure of total spontaneous activity (in Arbitrary Unit, UA; (**H**) and cumulative spontaneous activity (**I**) within 12 h for the 2 groups. ns: no significant difference; * *p* < 0.05, ** *p* < 0.01 and *** *p* < 0.001, significantly different from NaCl-treated mice.

**Figure 7 cells-09-02387-f007:**
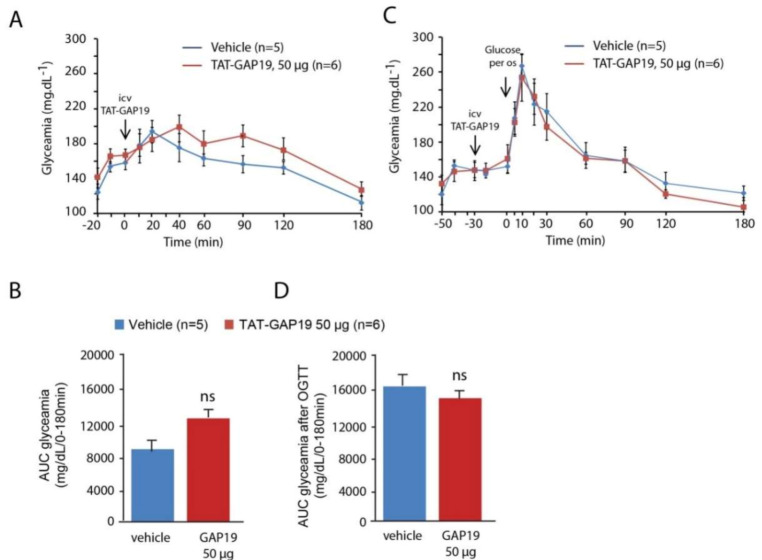
(**A**)**:** Glycaemia measured in fasted mice having received either i.c.v. injection of sodium chloride (NaCl) 0.9% or TAT-GAP19 50 µg. Arrow indicates the time of TAT-GAP19 icv administration. (**B**): Quantification of area under the curves (AUC) measured between 0 and 180 min after i.c.v injection of either NaCl 0.9% or TAT-GAP19 (50 µg). Arrows indicate the times of TAT-GAP19 and glucose administrations. (**C**)**:** Time-course changes in blood glucose level after per os glucose administration (1.5 g/kg) performed 30 min after i.c.v. injection of either NaCl 0.9% or TAT-GAP19 (50 µg) in fasted mice. (**D**): Quantification of AUC measured at between 0 and 180 min after per os glucose administration in mice having received either NaCl 0.9% or TAT-Gap19 (50 µg). ns: no significant difference.

**Figure 8 cells-09-02387-f008:**
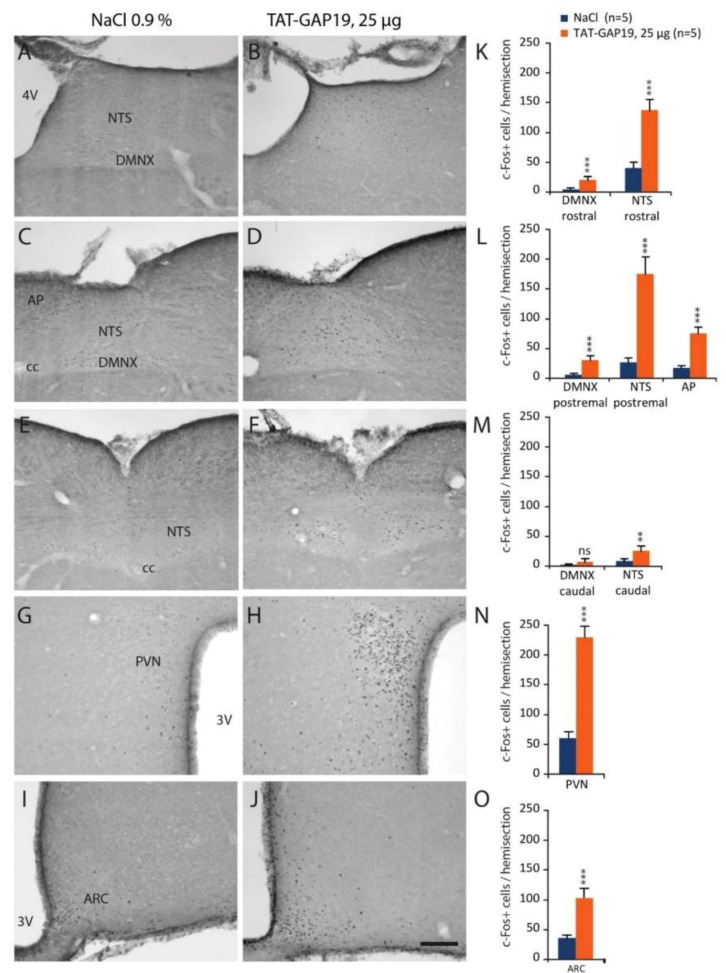
Microphotographs illustrating c-Fos protein immuno-labelling observed on coronal brain slices of mice treated with vehicle or TAT-GAP19 25 µg and sacrificed 3 h after injection. In response to TAT-GAP19 injection, c-Fos positive nuclei were located within the rostral (**A**,**B**), postremal (**C**,**D**) and caudal (**E**,**F**) dorsal vagal complex (DVC), the PVN (**G**,**H**) and the ARC (**I**,**J**). Scale bar: 200 µm. (**K**–**O**): Quantification of c-Fos positive cells in structures of interest for controls (*n* = 5) and for mice treated with TAT-GAP19 25 µg (*n* = 5). PVN: paraventricular nucleus, 3V: third ventricle, ARC: arcuate nucleus, 4V: fourth ventricle, NTS: nucleus tractus solitarius, AP: area postrema, cc: central channel. ns: no significant difference; ** *p* < 0.01 and *** *p* < 0.001, significantly different from vehicle-treated mice.

**Table 1 cells-09-02387-t001:** Primers sequences used for SYBR Green assays.

**Genes**		**Primer Sequences**	**Tm**	**T° qPCR**	**Amplicons**	**Genes Number**
GFAP	F	AGAGGAGTGGTATCGGTCTAAGTTT	56.9 °C	60 °C	167 pb	NM_001131020.1
	R	GCCGCTCTAGGGACTCGTTC	59.8 °C			NM_010277.3
Cx43 (mGja1)	F	AGAGCCCGAACTCTCCTTT	55.7 °C	60 °C	247 pb	NM_010288.3
	R	GTTCATCACCCCAAGCTGAC	56.1 °C			
Cx 30 (mGjb6)	F	CTGTCCCCGATTCCATTCCC	57.9 °C	60 °C	292 pb	NM_001010937.2
	R	CATCGTGCAGGCTTATTCTGAG	55.8 °C			
Cx 26 (mGjb2)	F	GCGACCCATTTCGGACCAA	58.2 °C	60 °C	159 pb	NM_008125.3
	R	GAGTGTGCCCCAATCCATCT	57.3 °C			
mPanx1	F	CAGGCTGCCTTTGTGGATTC	56.7 °C	60 °C	145 pb	NM_019482.2
	R	CGGGCAGGTACAGGAGTATG	57.2 °C			
mGAPDH	F	TTCTCAAGCTCATTTCCTGGTATG	54.9 °C	60 °C	143 pb	NM_008084.3
	R	GGATAGGGCCTCTCTTGCTCA	58.4 °C			

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
