# Peer review of "Blockade of Glial Connexin 43 Hemichannels Reduces Food Intake"

_cells, 2020, doi:10.3390/cells9112387_

Round 1

Reviewer 1 Report

These paper studied the role of astrocytic Cx43 hemichannels in mice food intake. These paper is well written, and logically designed. Results are very interesting. However, I have few comments, one of them that may require more experiments. 

Minors:

1.- In line 55, ..."hydrophilic channel of 2 nm in diameter"..please give a reference. 

2.- Figure 6A, The colour of TAT-gap19 mutated in the legend (purple) does not match with the TAT-GAP19 mutated bar (Gray)

Major:

In the whole work the authors constantly changed the concentration of TAT-GAP19..thus for example in figure 4A used 100 uM TAT-GAP19, but in 6A, they used 12.5 and 25 uM, why not used 100 uM as in figure 4A ? then in figure 5, they change again an used 50 uM. These constant changes in the concentration used are confusing and do not allow comparison of the different results obtained. Additionally, only in results 4A used the control TAT-GAP19 mutated, is also necesary to show the effect of this mutant in the other results, even more when there are constant changes in concentration. 

Author Response

We would thank the reviewer for the time spent evaluating our manuscript and his (her) positive and constructive feedback. The manuscript has been corrected taking into account most of the criticisms and suggestions raised.

Minors:

1.- In line 55, ..."hydrophilic channel of 2 nm in diameter"..please give a reference. 

Thank you for this comment. This reference has been added:  

Evans WH. Cell communication across gap junctions: a historical perspective and current developments. Biochem Soc Trans. 2015 Jun;43(3):450-9. doi:10.1042/BST20150056.

2.- Figure 6A, The colour of TAT-gap19 mutated in the legend (purple) does not match with the TAT-GAP19 mutated bar (Gray)

Sorry, this point has been corrected.

Major:

In the whole work the authors constantly changed the concentration of TAT-GAP19..thus for example in figure 4A used 100 uM TAT-GAP19, but in 6A, they used 12.5 and 25 uM, why not used 100 uM as in figure 4A ? then in figure 5, they change again an used 50 uM. These constant changes in the concentration used are confusing and do not allow comparison of the different results obtained.

Sorry for this confusing message, we have tried to clarify the text by adding the following information:

  • In vitro, we used TAT-GAP19 at the dose of 100 µm based on the pioneer work by Abudara et al (PMID: 25374505). This point has been added in the text.
  • Regarding the in vivo experiments, a dose response was performed and a reduction of food intake was observed from 12.5 µg with a maximal effect with 25 µg. This dose of 25 µg was used to perform i/ TAT-GAP19 diffusion after icv injection (Fig.5), food intake analysis (Fig. 6) and c-Fos staining (Fig.8).
  • We also check the specificity of the TAT-GAP19-induced anorexigenic effect, a range of 12.5-50 µg TAT-GAP19 was used and we tested a possible effect on energy expenditure, glycaemia, and locomotor activity. The absence of effect of TAT-GAP19 on these parameters was illustrated for the higher dose.

Additionally, only in results 4A used the control TAT-GAP19 mutated, is also necesary to show the effect of this mutant in the other results, even more when there are constant changes in concentration. 

We have checked the effects of GAP-19-I14A on food intake in Fig. 6A (not 4A), showing that GAP-19-I14A does not alter food intake at a dose of 25 mg/kg. Since GAP-19-I14A had no effect on food intake, we did not test its effect on c-Fos immunolabeling within the DVC and mediobasal hypothalamus, two structures strongly involved in food intake control.

Please note that GAP-19 displayed no effect neither on energy expenditure (Fig 6G-I), nor on OGTT (Fig. 7). Therefore, testing the effects of GAP-19-I14A on these parameters appeared unnecessary.

We have also checked this point on hypothalamic slices. TAT-GAP19 mutated (100 µm) did not inhibit BET uptake as TAT-GAP19 did (see the result on attached Fig. 1-Supp). However in hypothalamic slice experiments, we favored Cbx and TAT-GAP19 conditions, the TAT-GAP19 mutated peptide was not done enough times to provide quantification.

Reviewer 2 Report

The authors showed that pharmacological inhibition of CX43 reduces the food intake in mice. The authors suggest tonic delivery of orexigenic molecules related mechanisms regulating fasting and obesity.

Although the paper is of academic interest in an important area and presents novel effect, I have major concerns for the low-resolution images, limited experimental methods and insufficient mechanism related evidence.

First of all, there are problems with low-resolution images of immunostaining. The resolution of the image figures should be improved.

Not only is the image of low resolution, but the result of the image is quite ambiguous. What does this mean (ex; associated or related)? Isn't it colocalized? Methods such as z stack could be used for a more accurate interpretation of images.

There is a limitation to concluding the inhibiting effect of CX43 with only one pharmacological method. The authors can perform additional experiments using the acute gene silencing method or conditional knock out mice.

It is insufficient to discuss by just mention related references for the mechanism. The authors should show some experimental evidence proving a relationship between the orexigenic molecule and CX43.

Author Response

We thank the reviewer for the time spent evaluating our manuscript and his (her) constructive feedback and interesting discussion of ours results. The manuscript has been corrected taking into account most of the criticisms and suggestions raised.  Nevertheless, certain comments appear difficult to address due to the lack of reference to the figures/images pointed out. Anyway, we have answered the questions in a point-by-point fashion.

First of all, there are problems with low-resolution images of immunostaining. The resolution of the image figures should be improved.

Could the referee precise which images are to him (her) in low-resolution? All images provided are in TIFF format 300 dpi.

Please note that illustrations of voxel images (Fig.3) are at near the limit resolution of confocal microscopy and acquired with confocal parameters adjusted to minimize photobleaching. Thus, these images could appear as low resolution images.   

Not only is the image of low resolution, but the result of the image is quite ambiguous. What does this mean (ex; associated or related)? Isn't it colocalized? Methods such as z stack could be used for a more accurate interpretation of images.

One more time it’s difficult to answer this point and revise our Ms accordingly without clear reference to the concerned figure(s). Regarding Figure 1 (Cx43 and glial markers), we search here for the identity of Cx43-expressing cells. This point is mentioned in the Results section (Glial Cx43 expression in the hypothalamus and DVC).

“….Cx43 staining was found strongly associated with GFAP+ protoplasmic astrocytes from both structures (Fig. 1 B-G and H-I). A double staining of Cx43 with vimentin revealed the association of Cx43 with hypothalamic tanycytes (Fig. 1F) and tanycytes-like cells i.e. vagliocytes (Pecchi et al, 2007) of the DVC (not shown). Finally, co-staining of Cx43 with Ionized Ca2+ Binding Adapter Molecule 1 (IBA1), a marker of microglia, showed that Cx43 was sometimes found associated with these cells in the hypothalamus and DVC…”

In Figure 3, we looked at the juxtaposition between bassoon+ and Cx43+ structures. This point is extensively described in the Materials and Methods section (§ Stereological estimation of Cx43/Bassoon apposition) and in the Results section (Perisynaptic localization of Cx43).

We provide Z stack images for Figure 3 and reformatted this figure for a more precise interpretation of images.

 There is a limitation to concluding the inhibiting effect of CX43 with only one pharmacological method. The authors can perform additional experiments using the acute gene silencing method or conditional knock out mice.

We agree that Cx43 siRNA or knock-out could be interesting and useful but the question addressed with these approaches quite different from what we sought to address here. We choose to develop a pharmacological approach using Cx43 minetic peptide because it’s the unique possibility to target Cx43 hemichannels without interfering with Cx43 expression and subsequently with i/ Cx43-GAP junctions formation and ii/ communication through Cx43-GAP junctions.

It is insufficient to discuss by just mention related references for the mechanism. The authors should show some experimental evidence proving a relationship between the orexigenic molecule and CX43.

A this stage, as mentioned in the discussion, it seems difficult to specifically identify one (or several) orexigenic molecule(s) release by Cx43 hemichannels because most of the compounds transported through Cx43 hemichannels (ATP, glutamine, glutamate, aspartate, glycine, D-serine) have a profound impact on food intake. For instance, the use of icv injections of these molecules with orexigenic properties on animal pretreated with TAT-GAP19 will not provide convincing results regarding the nature of molecules endogenously released through Cx43 hemichannels.

Round 2

Reviewer 1 Report

The work has gained clarity, however I think the introduction needs to cite other works that show the role of hemichannels in the modulation of synapses, there are several works on the subject.

Minor comments:

line 51, separate that and Cx43.
Line 77, insert hemichannels before HCs
Line 84, delete Cx before GJ
Line 86, delete hemichannels before HCs, and delete the parentheses.
In results, there is an annoying mix of fonts, please be consistent in type and font size.

Reviewer 2 Report

The authors have responded appropriately to reviewer comments and enhanced the quality of the manuscript.